# SynthFormer: Equivariant Pharmacophore-based Generation of Molecules for Ligand-Based Drug Design

## Abstract

Drug discovery is a complex and resource-intensive process, with significant time and cost investments required to bring new medicines to patients. Recent advancements in generative machine learning (ML) methods offer promising avenues to accelerate early-stage drug discovery by efficiently exploring chemical space. This paper addresses the gap between in silico generative approaches and practical in vitro methodologies, highlighting the need for their integration to optimize molecule discovery. We introduce SynthFormer, a novel ML model that utilizes a 3D equivariant encoder for pharmacophores to generate fully synthesizable molecules, constructed as synthetic trees. Unlike previous methods, SynthFormer incorporates 3D information and provides synthetic paths, enhancing its ability to produce molecules with good docking scores across various proteins. Our contributions include a new methodology for efficient chemical space exploration using 3D information, a novel architecture called Synthformer for translating 3D pharmacophore representations into molecules, and a meaningful embedding space that organizes reagents for drug discovery optimization. Synthformer generates molecules that dock well and enables effective late-stage optimization restricted by synthesis paths.

## 1 Introduction

Drug discovery is a complex, lengthy and expensive process (Wouters et al., 2020). Computer aided drug design has been shown to be effective (Schneider & Fechner, 2005), while generative machine learning (ML) methods offer a promising way to speed up the early stages of drug discovery by exploring molecules in an efficient and cost-effective way (Meyers et al., 2021). There are two main computational methods for early-stage drug discovery: de novo ligand design and virtual screening. State-of-the-art ML methods for target based molecule generation, such as TargetDiff (Guan et al., 2023) and Pocket2mol (Peng et al., 2022), often overlook the practical aspects of synthesizing molecules in the lab. On the other hand, screening models like Equibind (Stärk et al., 2022) and Diffdock (Corso et al., 2023) are limited by the library that they need to screen. *In vitro* screening uses different methodologies. DNA encoded libraries (DEL) (Brenner & Lerner, 1992) and High-throughput screen (HTS) (Inglese et al., 2007) screening are costly, because they require the acquisition of large compound libraries, specialist equipment and use up large amounts of lab consumables. Dose-response assays (Crump et al., 1976) are limited by low throughput. Therefore, it is crucial to integrate *in vitro* technologies with *in silico* techniques to optimize molecule discovery, which will be specific for each target and disease to reduce cost and improve the speed of discovery.

For instance, DEL or dose-response screens can be enhanced with generative methods that suggest which molecules to screen, as each molecule would need to be ordered from a 3rd party supplier or made in house. These molecules are limited by the available building blocks and reactions (Fair et al., 2021). In contrast, HTS offers limited improvement potential from *in silico* methods, as its advantages stem from hardware optimization for a fixed setup and library. Consequently, not all combinations of ML and *in vitro* techniques provide acceleration; therefore, we should focus on identifying the right combination that can significantly enhance experimental outcomes.

One key failure point for generative ML methods in drug discovery is synthesisability. Most methods rely on the Synthetic Accessibility (SA) score. However, it has been demonstrated that the SA score (Ertl & Schuffenhauer, 2009) is not a discriminative feature and is unable to determine whether a molecule is synthesizable, as shown in Synflownet (Cretu et al., 2024). There have been multiple works that use a generative tool for a bottom up approach (Gao et al., 2022) (Cretu et al., 2024). However, none of these models incorporate 3D information.

Moreover, the most common molecule optimization approach relies on an external scorer blindly guiding the optimization as proposed by Guacamol (Brown et al., 2019) and Moses (Polykovskiy et al., 2018). However, in reality, the lead optimization process is heavily constrained by the available synthesis paths which also become the basis for patenting the Markush structure (Brown, 1991).

To address these challenges we propose the SynthFormer, a model that uses 3D equivarient encoder for pharmacophores and generates fully synthesisable molecules by building them as synthetic trees. We validate the effectiveness of the model by showcasing its ability to generate molecules that have good docking scores for a variety of proteins, as well as show that this tool can be effective for molecule optimisation. The main contributions of our work can be summarized as follows:

- We present a new methodology for encoding datasets to effectively learn and explore chemical space based on 3D information.
- We propose a novel architecture, called Synthformer, to translate 3D pharmacophore representations into molecules.
- We demonstrate that the Synthformer model is capable of generating molecules with strong docking performance.
- We show that the embedding space organizes reagents in a meaningful way, enabling more effective drug discovery optimization.
- We propose a simple yet effective optimization tool for later-stage optimization, constrained by synthesis paths.

## 2 RELATED WORK

**Combinatorial Optimization.** Approaches in combinatorial optimization employ a range of chemical building blocks and reaction templates to construct a combinatorial chemical space, such as Galileo (Meyenburg et al., 2023). These strategies then apply optimization techniques, such as genetic algorithms (Gao et al., 2021) and Monte Carlo tree search (Swanson et al., 2023), to explore this space for desired molecules. Prior to the advent of molecular deep learning, early efforts like SYNOPSIS (Vinkers et al., 2003) generated candidate molecules through virtual reactions, selecting them based on scoring functions. Recent advancements in deep learning for predicting chemical reactions (Coley et al., 2019) have further inspired methods that utilize neural networks to forecast reaction outcomes without relying on predefined templates.

**Synthesis based deep learning** Related work spans various approaches to molecule design, based on reinforcement learning (RL) or self-supervised learning. RL-based methods, such as SynFlowNet Cretu et al. (2024) and SynNet Gao et al. (2022), optimize molecular properties by simulating sequential decisions, constrained by synthesizability and reaction feasibility. The core limitation is that the reward function is of limited accuracy, the Synflownet model was guided with the docking module, which accuracy is 0.41. These methods rely on reward functions tailored to specific objectives, such as activity or docking scores. On the other hand, self-supervised learning approaches, like ChemProjector Luo et al. (2024), learn meaningful molecular representations by predicting structural or functional transformations in a molecule-to-molecule framework. However, this does not allow to effectively model activity cliffs Zhang et al. (2023), as it is only transforming the molecules based on the graph and has no binding information.

**3D generative models.** There are two primary approaches for shape-based generation: conditioning on a protein structure or on a known active ligand. Structure-based methods, such as TargetDiff (Guan et al., 2023) and Pocket2Mol (Peng et al., 2022), are designed to generate molecules that fit within a protein binding site. However, benchmark studies (Jocys et al., 2024) have demonstrated that while these methods can produce molecules that theoretically fit, they often fail to produce synthesis-ready compounds and may not bind as intended needing additional modifications as shown

in (Luo et al., 2024). Shape-based approaches, like SQUID (Adams & Coley, 2023) and LigDream (Skalic et al., 2019), that utilises a way to encode the shape or pharmacophores, and use another decoder to decode the molecules. Despite their promise, these networks have also struggled with generating synthesisable molecules.

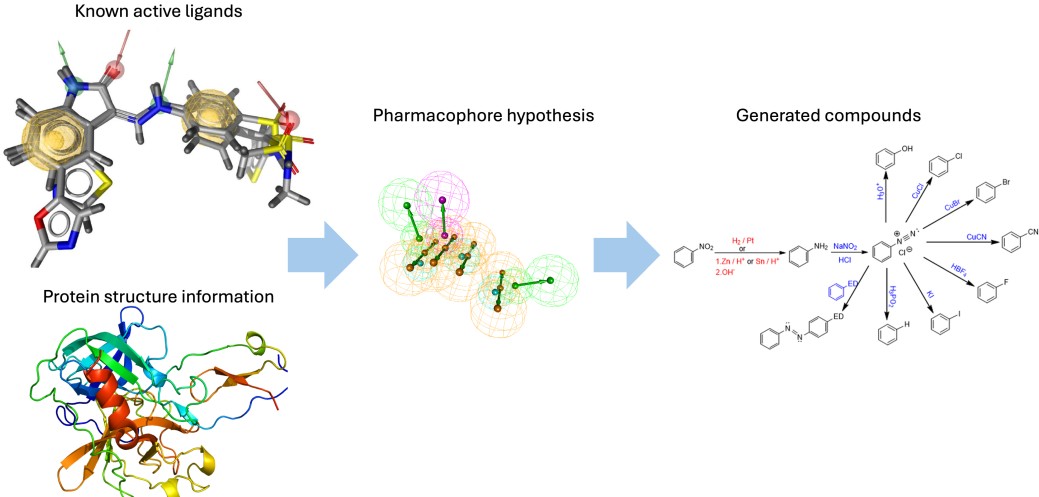

Figure 1: We take the available protein structure or information, along with data from active ligands, which can typically be used to extract a hypothesis. However, in our case, we focus on the crystallized ligand, extracting its pharmacophores, and then use these pharmacophores as the basis to generate new compounds. Images from Wikipedia (n.d.); Seidel et al. (2010); Prep (2024)

**Molecule Optimization** in a machine learning context was described in Guacamol (Brown et al., 2019) and Moses (Polykovskiy et al., 2018) as a process where molecules are improved by using scoring functions to evaluate how well they fulfill desired property profiles. The goal is to generate molecules that maximize these scores, while separating this optimization process from the challenge of selecting the appropriate scoring functions (Schneider & Fechner, 2005). This approach has been used in the ML community for SMILES based representation (Cao & Kipf, 2022), Graph based representation without 3D information, RL approaches (Cretu et al., 2024), as well as 3D approaches (Peng et al., 2022). At the same time, this optimization framing is detached from the drug discovery process in real life. In drug discovery, once HITs are identified and Structure-activity relationship is established, lead optimization involves exploiting the known synthetic pathways and exploring similar routes to improve the molecule. This was done in this lead optimization study (Harris & Faucher, 2019), where they explain how they went about finding and optimizing 4,5-Dihydropyrazoles as Mono-Kinase Selective, Orally Bioavailable and Efficacious Inhibitors of Receptor Interacting Protein 1 (RIP1) Kinase.

**Limitations of current state-of-the-art.** Combinatorial optimization techniques are constrained by their reliance on predefined reaction templates, which result in a slow exploration of chemical space. On the other hand, deep learning models typically focus on only one part of the task: generating structurally and physically accurate structures or synthetically accessible compounds, but not both. Moreover, while deep learning methods excel in these areas, they often fail to integrate the optimization process with the specific needs of lead optimization. As a result, the generated compounds may not align well with the practical requirements of drug discovery, such as feasibility of synthesis or sufficient solubility for assay conditions. This detachment between structural design and practical application can hinder the overall success of deep learning approaches in optimizing drug candidates.

# 3 METHOD

## 3.1 MOLECULE REPRESENTATION

**Chemical space** A chemical space $S$ is a set of molecules generated by a finite set of building block molecules $B = \{B_1, B_2, \ldots, B_N\} \subset S$ and a set of reaction rules $R = \{R_1, R_2, \ldots, R_M\}$. Formally, a reaction rule maps reactants to a reaction's product. For example, a reaction rule $R$ with two reactants can be denoted by:

$$R : B_1 \times B_2 \to S$$

$$(B_1, B_2) \mapsto Y,$$

where $B_1$ and $B_2$ are sets of molecules to which reaction $R$ can be applied, and $Y$ is the *main reaction product*. The chemical space is generated by starting from the building blocks and iteratively applying the reactions to every possible combination of molecules. There are some cases where a reaction has multiple possible main products. For ease of discussion, we assume that reactions produce only one main product in this section.

Every molecule in the chemical space is a product of a synthetic process, which involves applying reaction rules to building blocks and intermediate molecules to build a complex molecule. For example, $P2 = R_2(P1, B_3)$, where $P1 = R_1(B_1, B_2)$ denotes a process in which we first apply reaction $R_1$ to building blocks $B_1$ and $B_2$, and then apply reaction $R_2$ to building block $B_3$ and the product of $R_1(B_1, B_2)$.

**Conformers** A molecule conformer is a spatial arrangement of atoms in a molecule that can be rotated around single bonds. Consider a molecule as a graph $G = (\mathcal{V}, \mathcal{E})$ with atoms $v \in \mathcal{V}$ and bonds $e \in \mathcal{E}$, and denote the space of its possible conformers $\mathcal{C}_G$. A conformer $C \in \mathcal{C}_G$ can be specified in terms of its *intrinsic* (or internal) coordinates: local structures $L$ consisting of bond lengths, bond angles, and cycle conformations; and torsion angles $\tau$ consisting of dihedral angles around freely rotatable bonds (precise definitions in Appendix A). For each molecule, the coordinates can be represented as a set $\{\mathbf{x}_v^k \in \mathbb{R}^3 \mid v \in \mathcal{V}, k \in \{1, 2, \ldots, n_v\}\}$, where $\mathbf{x}_v^k = (x_v^k, y_v^k, z_v^k)$ represents the $k$-th set of 3D coordinates for atom $v$, and $n_v$ is the number of possible solutions for the coordinates of atom $v$.

**Pharmacophores** are abstract representations of the molecular features necessary for molecular recognition of a ligand by a biological macromolecule, crucial for its biological activity. For each atom of the molecule that contains a pharmacophore we assign a one hot encoding. The one-hot encoding vector $\mathbf{p} = [p_1, p_2, p_3, p_4, p_5, p_6]$ for the pharmacophore types available in the RDKit (Landrum, 2006) package represents the presence (1) or absence (0) of Hydrogen Bond Donors (HBD), Hydrogen Bond Acceptors (HBA), Aromatic Rings, Hydrophobic Centers, Positive Ionizable, and Negative Ionizable features, respectively.

**Similarity scores** To evaluate the similarity between the input molecule and the output molecule, we use the Tanimoto similarity score on three different fingerprints: (1) Morgan fingerprint of length 4096 and radius 2 (Morgan, 1965), (2) Morgan fingerprint of Murcko scaffold (Bemis & Murcko, 1996), and (3) Gobbi pharmacophore fingerprint (Gobbi & Poppinger, 1998). The three similarity scores indicate chemical similarities in three different aspects: overall structure, scaffold structure, and pharmacophore property, respectively, and are all normalized to [0, 1].

**Equivariant pharmacophore encoder** In our approach, we represent pharmacophores through a one-hot encoding scheme. Specifically, we denote a pharmacophore feature vector as $\mathbf{f} = [f_1, f_2, \ldots, f_n]$, where each $f_i \in \{0, 1\}$ and $\sum_{i=1}^n f_i = 1$. This representation allows us to capture the presence or absence of distinct pharmacophoric features in a binary format. Additionally, we employ Cartesian coordinate features to preserve equivariance. We treat this as a fully connected graph, because our hypothesis space is not limited by any bonds. We use the EGNN (Satorras et al., 2021), that takes as input the set of node embeddings $\mathbf{h} = \{\mathbf{h}_0, \ldots, \mathbf{h}_{M-1}\}$, coordinate embeddings $\mathbf{x}' = \{\mathbf{x}_0', \ldots, \mathbf{x}_{M-1}'\}$, and edge information $\mathcal{E} = (e_{ij})$ and outputs a transformation on $\mathbf{h}^{l+1}$ and $\mathbf{x}'^{l+1}$. The operation $\mathbf{h}^{l+1}, \mathbf{x}'^{l+1} = \text{EGCL}(\mathbf{h}, \mathbf{x}', \mathcal{E})$ denotes an equivariant graph convolutional layer (EGCL), a specific variant of equivariant neural networks (EGNNs) as introduced in Satorras et al. (2021). The equations that define this layer are the following:

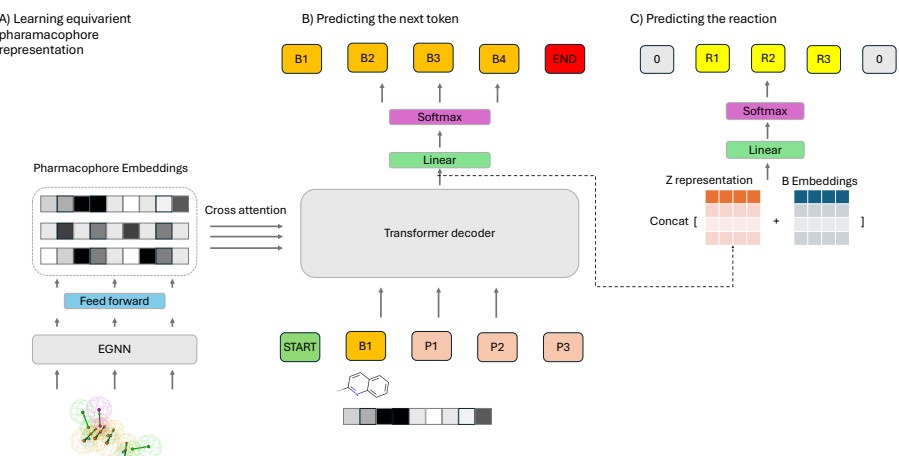

Figure 2: The generation process begins by representing the pharmacophores as a fully connected graph, passed through a GNN to obtain pharmacophore embeddings. These embeddings are then propagated to a transformer decoder. The transformer decoder first receives a start token and predicts building block (B1), followed by reaction 0. Next, it takes the fingerprint of B1 as input to the decoder and predicts building block 2 (B2), followed by reaction 1 (R1), which is applied to generate product 1 (P1). Product 1 is then used to predict building block 3 (B3) and reaction 2 (R2), producing product 3(P3). This process repeats until the end token is generated.

$$m_{ij} = \phi_e\left(\mathbf{h}_i^l, \mathbf{h}_j^l, \left\|\mathbf{x}_i' - \mathbf{x}_j'\right\|_2^2, a_{ij}\right) \tag{1}$$

$$\mathbf{x}_i'^{l+1} = \mathbf{x}_i' + \sum_{j \neq i}(\mathbf{x}_i' - \mathbf{x}_j')\phi_x(m_{ij}) \tag{2}$$

$$m_i = \sum_{j \in \mathcal{N}(i)} m_{ij} \tag{3}$$

$$\mathbf{h}_i^{l+1} = \phi_h\left(\mathbf{h}_i^l, m_i\right) \tag{4}$$

In equation 1, we input the relative squared distance between two coordinates $\left\|\mathbf{x}_i' - \mathbf{x}_j'\right\|_2^2$ into the edge operation $\phi_e$. Later, we use the representation $\mathbf{h}^{l+1}$ and pass it through cross-attention to the decoder. We employ a seven-layer EGNN as the encoder in our model.

**Synthetically accessible molecule decoder** The input to the transformer begins with a token [START], followed by first B, then P1. The input represents the last final product from the reactions. The model stops when the building block MLP predicts the end token [END]. An input token is a morgan fingerprint and start token , denoted by $[\texttt{fp}, start]$, where $f_p \in \{0,1\}^{1024}$ is the Morgan fingerprint of length 1024 and radius 2 (Morgan, 1965). Building block tokens are converted into embeddings by an MLP. Moreover we use positional encodings to encode the position for each token.

$$\mathbf{h}_i^{(0)} = \text{PE}(i) + \begin{cases} \boldsymbol{\epsilon}_{\text{start}}, \text{type}(T_i) = \text{START} \\ \text{MLP}_{\text{fp}}(p_i), \text{type}(T_i) = \text{B} \end{cases} \tag{5}$$

The decoder uses the standart multihead attention mechanism. However, at the output of the decoder, we take the multihead output pass and concatenate with the learnable molecular embeddings.

$$\text{MultiHead}(Q, K, V) = \text{Concat}(\text{head}_1, ..., \text{head}_h)W^O \tag{6}$$

$$\text{where head}_i = \text{Attention}(QW_i^Q, KW_i^K, VW_i^V), \tag{7}$$

where the projections are parameter matrices $W_i^Q \in \mathbb{R}^{d_{\text{model}} \times d_k}$, $W_i^K \in \mathbb{R}^{d_{\text{model}} \times d_k}$, $W_i^V \in \mathbb{R}^{d_{\text{model}} \times d_v}$, and $W^O \in \mathbb{R}^{h d_v \times d_{\text{model}}}$. The encoder is composed of a stack of $N = 7$ identical layers and we employ $h = 8$ attention heads. As we pass the input tokens through the last multi-head attention mechanism to obtain the representation $Z$. This representation serves as the basis for predicting the next building block $B_{i+1}$, using a softmax applied to a linear transformation of $Z$, also refered as MultiHead(Q, K, V). The corresponding equation is given by:

$$B_{i+1} \sim \text{Softmax}(\text{Linear}(Z_i)) \tag{8}$$

Next, we concatenate the representation $Z_i$ with the embeddings of the predicted building block $B_{i+1}$ and apply a linear transformation followed by a softmax function to predict the reaction $R_{i+1}$. This process can be expressed as:

$$R_{i+1} \sim \text{Softmax}(\text{Linear}(\text{Concat}(Z_i, \text{Embedding}(B_{i+1})))) \tag{9}$$

The model uses both the predicted building block $B_{i+1}$ and the reaction $R_{i+1}$ to generate the product, which is then fed back into the model as input for the next iteration. This process continues iteratively, generating new building blocks and reactions, until the model predicts an end token for the building block, signaling the completion of the process.

## 3.2 TRAINING

In our study, we employ causal masking within the transformer architecture to ensure that the model adheres to the autoregressive property during training and inference. Causal masking is crucial for preventing the model from accessing future tokens in the sequence when predicting the current token, thereby maintaining the integrity of the temporal or sequential data flow.

The building block loss function for building block fingerprint where $\ell$ is the length of the sequence :

$$L_{\text{b}} = \frac{1}{n} \sum_{j=0}^{l-1} \text{CE}(\hat{B}_{i+1}, B_{i+1}), \tag{10}$$

where CE is the cross entropy loss function, for reaction type:

$$L_{\text{rxn}} = \frac{1}{m} \sum_{i=0}^{\ell-1} \text{CE}(\hat{r}_{i+1}, r_{i+1}). \tag{11}$$

The final loss function is the sum of the two terms: $L = L_{\text{bb}} + L_{\text{rxn}}$.

## 3.3 INFERENCE

At inference time, we prepare the pharmacophore data and the start token as inputs. We generate tokens one by one. The transformer decoder first receives a start token and predicts building block B1, followed by reaction 0. Next, it takes the fingerprint of B1 as input and predicts B2, followed by reaction 1, which is applied to generate product 1. Product 1 is then used to predict building block 3 and reaction 2, producing product 3. This process repeats until the end token is generated. Finally, an end token [END] terminates the process and the last molecule is returned as the product.

## 4 EXPERIMENTAL SETUP

### 4.1 DATASETS

**Reaction templates** We use the publicly available reaction template set (Hartenfeller et al., 2012). The template set contains 58 reaction templates encoded in SMARTS string format.

| PDBID | Ref Dock | Average Dock Gen | Min Dock Gen | Tanimoto | Murcko | Gobbi |
|-------|----------|------------------|--------------|----------|--------|-------|
| 1x8d  | -6.17    | -6.32 ± 0.49     | -8.39        | 0.06     | 0.06   | 0.20  |
| 1xbo  | -10.79   | -7.22 ± 1.08     | -10.07       | 0.12     | 0.12   | 0.39  |
| 2afw  | -3.85    | -7.89 ± 0.91     | -9.13        | 0.10     | 0.07   | 0.38  |
| 2aog  | -10.45   | -8.75 ± 1.24     | -11.62       | 0.07     | 0.00   | 0.30  |
| 2bt9  | -6.68    | -6.73 ± 0.62     | -9.23        | 0.04     | 0.02   | 0.21  |
| 3coy  | -12.41   | -7.14 ± 4.36     | -11.26       | 0.10     | 0.08   | 0.43  |
| 3ga5  | -9.68    | -3.33 ± 1.61     | -5.60        | 0.05     | 0.03   | 0.23  |
| 4q6d  | -7.41    | -7.53 ± 1.54     | -10.07       | 0.07     | 0.10   | 0.32  |
| 5fl4  | -8.09    | -7.52 ± 0.83     | -9.28        | 0.10     | 0.10   | 0.37  |
| 5ka1  | -7.67    | -6.74 ± 1.07     | -9.05        | 0.07     | 0.06   | 0.42  |

Table 1: The table shows reference docking energies (Ref Dock) and the mean and minimum docking energies (Dock Gen) for 100 generated molecules docked to the same protein. Tanimoto, Murcko, and Gobbi similarities indicate the averaged structural resemblance between the 100 generated molecules and the reference ligands.

**Building blocks** We use the building blocks from Mcule (Mcule Team, 2023). The building blocks that fail the RDKit sanitization check or do not match any reaction template are removed. Duplicate building blocks are also removed. The pre-processing procedure leads to the final building block set containing 10858 molecules.

## 4.2 DATA GENERATION

To generate the data, we begin by selecting a set of molecules and perform a sequence of $n$ random reactions from the available set, saving the resulting products. Next, we repeat the process but include the option to sample both the initial molecules and the previously generated products, effectively exploring the chemical space through random sampling. Once the molecules are sampled, we compute a random conformer for each one using Merck Molecular Force Field (MMFF) optimization from RDKit (Landrum, 2006). Afterwards, we extract the pharmacophores and 3D coordinates of each molecule. This results in an input dataset where the pharmacophores and 3D coordinates serve as the inputs, and the output prediction corresponds to the synthetic tree.

## 5 RESULTS

### 5.1 DESIGNING ACTIVE COMPOUNDS

We evaluate the model by selecting 10 compounds from different PDB entries and extract their pharmacophores. These pharmacophores are used as inputs to generate 100 new compounds per original ligand. We assess the structural similarities between the generated and original compounds, followed by redocking the generated compounds. The docking scores are then compared to those of the original ligands to evaluate how closely the generated compounds mimic the binding affinity and pose.

In this study, we evaluated the structural similarities between 100 generated molecules (See Appendix A Figure 4) and reference ligands for various PDB entries. The docking results reveal a range of binding affinities for the generated molecules, with mean docking energies varying from -3.33 to -8.75 kcal/mol (see Table 1 column 3). Notably, we can see that for the PDB entries 1x8d, 2bt9, 416d, 5fl4 the average docking energy and the reference docking energy are extremely close, while the smallest docking energy is always lower than the reference docking energy, except for 3ga5.

The calculated similarity metrics—Tanimoto, Murcko, and Gobbi—provide insights into the structural resemblance between the generated molecules and reference ligands. Tanimoto and Murko similarities ranged from 0.03 to 0.12 showing that the similarity is extremely low. On the other hand, Gobbi similarities ranged from 0.20 to 0.43, showing the 2d Pharmacophore similarity is greater, but this is far from being identical. Thus, these results highlight that this model is able to maintain 2D pharmacophore similarity better than any kind of fingerprint similarity. Moreover, the Gobbi similarity is a 2D similarity and we prompt the model with 3D representations, explaining why the similarity scores are not closer to 1.

| PDB ID | MW (g/mol) | LogP 5% | LogP 95% | QED |
|--------|-----------|---------|----------|-----|
| 1x8d | 320.25 | 0.27 | 5.14 | 0.59 |
| 1xbo | 313.36 | 0.00 | 4.09 | 0.60 |
| 2afw | 320.32 | -0.01 | 5.41 | 0.58 |
| 2aog | 320.53 | 0.63 | 4.73 | 0.60 |
| 2bt9 | 305.96 | 0.02 | 4.60 | 0.60 |
| 3coy | 313.24 | 0.12 | 4.72 | 0.59 |
| 3ga5 | 318.71 | 0.24 | 5.13 | 0.60 |
| 4q6d | 331.75 | -0.10 | 5.23 | 0.57 |
| 5fl4 | 319.19 | 0.14 | 4.77 | 0.58 |
| 5ka1 | 339.02 | -0.41 | 4.80 | 0.56 |

Table 2: Properties of the generated molecules, featuring molecular weights of the generated molecules(MW) ranging from 305.96 g/mol to 339.02 g/mol, indicating a diverse array of molecular sizes. LogP values fall within the expected range according to the Ghose filter (typically -0.4 to 5.6), and we obtain reasonable QED scores that reflect the drug-likeness of the molecules.

We evaluate the generative model by selecting 10 molecules from PDB Bind and generate 100 compounds for each. The evaluation focuses on two key aspects. First, we perform a similarity analysis, comparing the generated molecules to the original ones by calculating similarity scores using Morgan, Murcko and Gobbi fingerprints. Second, we analyze the range of physicochemical properties, such as logp, druglikeness, synthetic accessibility, to assess the diversity and novelty of the generated compounds. This analysis helps gauge the relevance and diversity of the generated molecules, ensuring they maintain desirable properties while exploring the new chemical space.

## 5.2 EMBEDDING EXPLORATION

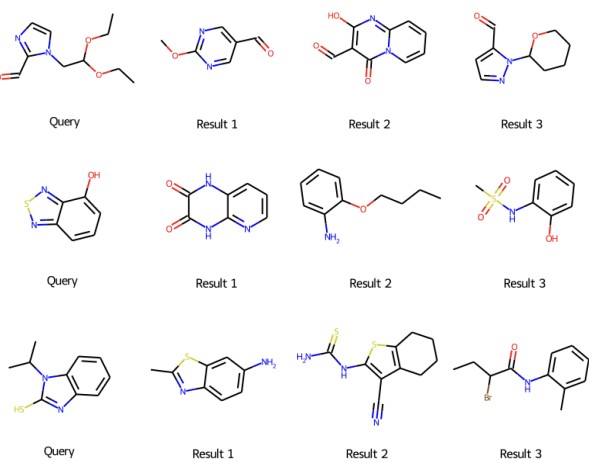

Figure 3: The embedding of the selected query molecule is used as the reference, and the three closest molecules are identified using cosine similarity for qualitative assessment. As observed, these molecules preserve significant structural similarity to the reference.

To further investigate the structural similarity between building blocks, we leveraged the embeddings produced by our transformer model. Specifically, we extracted these embeddings and employed cosine similarity to quantify the closeness of different building blocks.

Cosine similarity is a measure of the angular distance between two vectors, providing insight into their relative orientation in the embedding space. By applying this metric to the embeddings, we identified pairs of building blocks that are closely aligned according to the model's learned representations.

Our analysis revealed that the building blocks exhibited a high degree of similarity, particularly in terms of preserving ring structures and the number of nitrogen atoms. The embeddings captured

these structural features with remarkable fidelity, allowing us to observe that similar building blocks not only shared comparable ring configurations but also maintained consistent nitrogen counts.

Moreover, we took 100 building blocks sampled 10 most similar building blocks based on cosine similarity and computed the max similarity. 30% of building blocks had an average of 0.67 Tanimoto similarity. This result underscores the effectiveness of the transformer model in encoding structural nuances and highlights its capacity to differentiate and preserve essential chemical features within the embedding space. The preservation of rings and nitrogen counts suggests that the model has effectively learned to represent these critical components of molecular structures, leading to meaningful and accurate similarity measures.

### 5.3 Property Optimisation through genetic embedding modifications

We implement a genetic algorithm approach (See Appendix B) for molecular optimization within a synthetic reaction tree framework, aiming to discover molecules with improved properties. In this part we enhance molecular properties by altering reagents in the reaction steps within the known domain of the hypothesis. For each of the 10 molecules in the Protein Data Bank (PDB), we apply these modifications, using cosine similarity to adjust the reagents in the predicted list to obtain the desired molecules and we resample the reactions. This process helps identify pathways that yield the most optimal molecules based on targeted properties.

Once modifications are applied, we re-dock the resulting molecules, compare their properties, such as binding affinity and physicochemical properties (e.g., logP, druglikeness), and analyze the distributions. By iterating through cycles of modification and evaluation, we quantify improvements in properties over 3 cycles. We generate new molecules score them pick the top molecules and repeat this process 3 times. We show that we can decrease the logp by 0.21 and for a group of molecules druglikeness increase 0.09, while the average energy increases by 0.03 kcal/mol (worse).

### 5.4 Hit Expansion

Our model is also effective for hit expansion in drug discovery, providing a strategic method for identifying structurally similar and synthesizable analogs of hit molecules (Keseru & Makara, 2006; Levin et al., 2023). To demonstrate this, we applied our model to design inhibitors for all 10 proteins that have been used for the active ligand design task, following the setup from Levin et al. (2023).We constrained the generation process by initializing with the original structure and then sampling further for possible modifications. Our objective was to identify potentially active synthesizable molecules that are close to the molecule from the crystal structure.

To achieve this, we encode each crystallized molecule as a Morgan fingerprint and input the [Start] token along with the Morgan Fingerprint. We then sample an additional building block and the relevant reactions necessary to grow the molecules. A total of 100 molecules are sampled (see Appendix A Figure 5) and docked back to the relevant protein structures. Afterward, we select the pose with the best score and compare it against the score of the reference ligand.

We generated 100 analogs for each seed molecule, resulting in a diverse set of molecules with varying scores and structural similarities. Impressively, 11.4% of these analogs surpassed the original hit in terms of energy, while maintaining an average Tanimoto similarity of 0.67.

## 6 Conclusions

The evaluation of the generative model demonstrates its capacity to generate chemically diverse and relevant compounds while maintaining desirable physicochemical properties. By employing a variety of similarity metrics—such as Tanimoto, Murcko, and Gobbi fingerprints—and analyzing structural properties, the generated molecules are shown to maintain low similarity to reference ligands, while preserving pharmacophore information. Furthermore, the genetic algorithm-based optimization effectively guides the exploration of new chemical spaces, improving molecular properties while preserving scaffold integrity. The embedding analysis further validates the model's ability to capture key molecular features, such as ring structures and nitrogen atoms, illustrating its proficiency in maintaining meaningful chemical relationships. These results indicate that SynthFormer has strong potential to generate novel, drug-like molecules that can support hit identification, hit expansion and lead optimization efforts in drug discovery.

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

# A APPENDIX

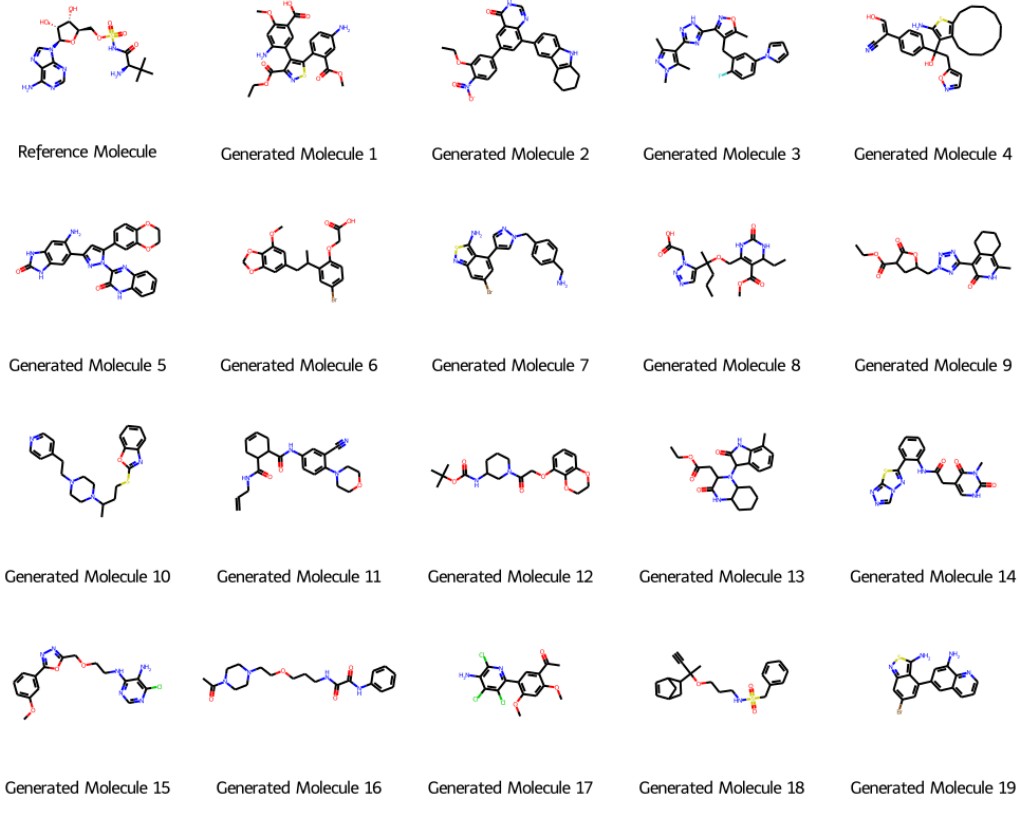

Figure 4: The first molecule corresponds to the reference structure derived from the 3COY PDB ID. The subsequent molecules are computationally generated using the Synthformer model, illustrating its capability to design novel molecular structures based on a known protein-ligand complex. These results highlight Synthformer's potential in generating diverse and plausible molecular candidates.

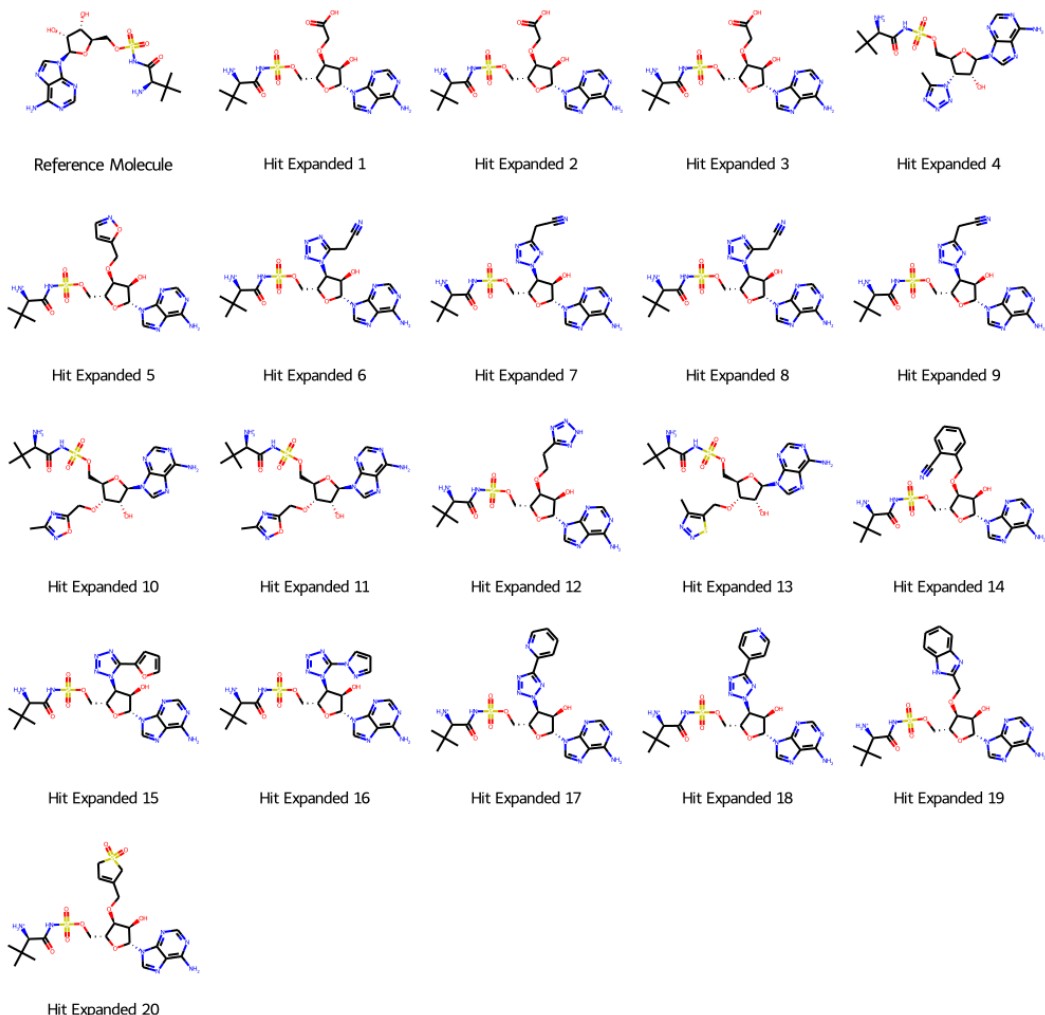

Figure 5: The first molecule corresponds to the reference structure derived from the 3COY PDB ID. The subsequent molecules are expanded hits generated by the Synthformer model, demonstrating structural diversity and novel chemical scaffolds.

# B    GENETIC ALGORITHM

---

**Algorithm 1** Synthetic Tree Generation and Scoring Algorithm

---

**Require:** Set of building blocks $\mathcal{B}$, reaction rules $\mathcal{R}$, scoring function $S(\cdot)$, number of iterations $T = 3$, and top-$k$ selection size $k$
1: Initialize the synthetic tree $\mathcal{T} \leftarrow \emptyset$
2: Initialize the set of molecules $\mathcal{M}_0 \leftarrow \mathcal{B}$
3: **for** $t = 1$ to $T$ **do**
4:     $\mathcal{M}_{\text{new}} \leftarrow \emptyset$
5:     **for** each molecule $m \in \mathcal{M}_{t-1}$ **do**
6:         **for** each building block $b \in \mathcal{B}$ **do**
7:             Find the nearest building block $b_{\text{nearest}} \in \mathcal{B}$ for $b$
8:             Apply reaction rules $\mathcal{R}$ to $m$ and $b_{\text{nearest}}$ to generate new molecule $m_{\text{new}}$
9:             Add $m_{\text{new}}$ to $\mathcal{M}_{\text{new}}$
10:         **end for**
11:     **end for**
12:     Score all molecules in $\mathcal{M}_{\text{new}}$ using $S(\cdot)$
13:     Select top-$k$ molecules from $\mathcal{M}_{\text{new}}$ based on scores to form $\mathcal{M}_t$
14:     Update the synthetic tree $\mathcal{T} \leftarrow \mathcal{T} \cup \mathcal{M}_t$
15: **end forreturn** Synthetic tree $\mathcal{T}$ containing all generated molecules

---

