# OpenReview forum: "SynthFormer: Equivariant Pharmacophore-based Generation of Molecules for Ligand-Based Drug Design"
_ICLR.cc/2025/Conference — Submitted to ICLR 2025_

### Official Review · Reviewer_TJa9 · 2024-10-16

**Soundness:** 1
**Presentation:** 1
**Contribution:** 2
**Rating:** 3
**Confidence:** 4

**Summary:**

This paper proposes SynthFormer for ligand-based drug design. Specifically, the authors highlight the challenge of synthesizability, and argue that Synthformer benefits from synthetic trees that incorporate pharmacophore and reaction information, addressing the gap between in-silico generative methods and in-vitro needs. A set of 10 proteins are adopted to showcase the results of SynthFormer.

**Strengths:**

The authors explored the potential of pharmacophores in generating synthesizable molecules via synthetic trees, and developed a way to utilize the reaction templates as well as other building blocks for synthetic tree data generation.

**Weaknesses:**

Major:
- **Presentation**. The organization of this paper could be improved. Some experimental setup (most of 3.2 data preparation) is mixed into the method section. The authors could consider moving the data preparation details to a separate "Experimental Setup" section, and elaborating other implementation details necessary for reproducibility (see Questions).
- **Unclear Scope**. In the abstract and introduction, this paper claims to "addresses the gap between in silico generative approaches and practical in vitro methodologies". If I'm not mistaken, the former refers to (1) molecular generative models, and (2) screening methods such as EquiBind and DiffDock, while the latter means relatively costly DEL and HTS. The authors then place a particular emphasis on synthesizability for generative ML methods, and elaborate on how it's been tackled afterwards, where the screening seems irrelavant and becomes out of topic. Could the authors explicitly state which aspects of the in silico - in vitro gap they are addressing, and how their focus on synthesizability relates to this gap? Better coordination would help to make a concrete statement about the research problem.
- **Motivation Not Justified**. It seems that this paper is motivated by the lack of incorporating truly synthetic paths into molecule generation. However, the performance of the proposed SynthFormer in proposing more synthesizable molecules remains to be justified in the experiment results, where only some molecular properties (docking scores, QED, MW and LogP) and molecular similarities are adopted as metrics. The authors argue that SA score is not a good metric, but no other metrics are utilized to validate their performance. In this case, it would be better if the authors could introduce more reliable metrics as the indicator of synthesizability, for example using retrosynthesis planning tools or expert chemist assessments. If this is not immediately possible, they might as well report SA instead to support the claim.
- **Lacking Comparison with Baselines**. This paper is not the first in exploring synthesizability for ligand-based drug design.  For other ligand-based methods that focus on synthesizability, methods like [1][2] are simply discarded by the authors as they are not 3D models, but a comparison will be needed to justify the authors' choice of incorporating 3D information and to show the significance. Moreover, recent works like [3] are missing in the discussion. The authors would need to make a detailed comparison with those counterparts so as to highlight their contribution and novelty.
- **Limited Evaluation**. The authors conducted four kinds of experiments, without any comparison to other generative models nor optimization baselines. It would be more convincing if the authors consider adding these comparisons: (1) For 4.1 designing active compounds and 4.4 hit expansion, generative models in structure-based drug design (SBDD), such as Pocket2Mol and TargetDiff as mentioned by the authors, could be included for a comparison between ligand-based drug design (LBDD) and SBDD, especially in terms of similarity and synthesizability. The performance of optimization approaches given reference ligands as seed molecules can also be shown here [4][5]. (2) For 4.3 property optimization, the authors only evaluate SynthFormer across 10 compounds in PDB, undermining the reliability of their results. As a ligand-based approach, the authors could conduct experiments on ChEMBL or other large datasets for a comprehensive and credible evaluation. Moreover, when it comes to optimizing binding affinities, the authors could consider comparing the optimization performance and efficiency with other optimization methods like RetMol [4], RGA [5] and DecompOpt [6].

Minor:
- Please remain consistent with the capitalized model names, such as Pocket2Mol / Pocket2mol, and Synthformer / SynthFormer.
- Caption of Figure 1 does not end with a period.
- The authors might as well consider using \paragraph instead of \textbf for those in related works and method sections for better readability.

[1] Amortized tree generation for bottom-up synthesis planning and synthesizable molecular design. https://arxiv.org/abs/2110.06389

[2] Synflownet: Towards molecule design with guaranteed synthesis pathways. https://arxiv.org/abs/2405.01155.

[3] Projecting Molecules into Synthesizable Chemical Spaces. https://arxiv.org/abs/2406.04628

[4] Retrieval-based Controllable Molecule Generation. https://arxiv.org/abs/2208.11126

[5] Reinforced Genetic Algorithm for Structure-based Drug Design. https://arxiv.org/abs/2211.16508

[6] DecompOpt: Controllable and Decomposed Diffusion Models for Structure-based Molecular Optimization. https://arxiv.org/abs/2403.13829

**Questions:**

- How does the genetic algorithm for molecular optimization within such synthetic reaction tree framework work? Could the authors provide the algorithm, as well as the experimental details in the Appendix?
- Why is 3D information necessary here in synthesizable ligand generation? Can the authors provide detailed structural evaluation to shed light on the importance of 3D modeling here, e.g. the distributions of rings, bond lengths, bond angles, and torsion angles?

---

> ### Author Response · Authors · 2024-11-23
>
> We thank the reviewer for their comments, and we provide detailed responses to each specific critique below.
>
> 1. Thank you for your suggestion. As requested, we have reorganized the paper by moving the data preparation details from the method section to a separate "Experimental Setup" section. We have also included additional implementation details to ensure better clarity and reproducibility.
>
> 2. We appreciate your point about the potential confusion regarding the scope. To clarify, the paper aims to address the gap between in silico generative methods and practical in vitro screening techniques, such as DEL and HTS, by considering both aspects in a holistic way. In silico methods, such as generative models and screening methods like EquiBind and DiffDock, are often limited in terms of their applicability to specific targets or reaction types. For example, DEL can only be used for certain targets and reactions, and HTS may have specific conditions or limitations.
> Synthformer bridges this gap by enabling the generation of synthesizable molecules based on pharmacophore information, while also taking into account available reactions and building blocks. This allows for the creation of libraries that are optimized for a given screening method and target, providing a more flexible and comprehensive approach to drug discovery.
> We have updated the manuscript to make this connection clearer, outlining how Synthformer can facilitate both in silico exploration and practical in vitro validation, by tailoring molecular generation to the specifics of available reactions and targets.
>
> 3. As stated in the paper, all molecules generated by Synthformer are synthesizable, as we focus on rule-based exploration, where we generate molecules based on predefined, synthesis-ready reactions and building blocks. This approach ensures that the molecules we propose are synthesizable by design.
> Regarding the use of the SA score, we acknowledge that it is a commonly used metric, but as demonstrated by SynFlowNet, it is not a reliable predictor of synthesizability, as it produces inconsistent results when applied to large molecule libraries, such as the ZINC database. Thus our model only generates molecules that can be synthesised by the available chemistry and using another score to assess synthesisability does not shed any additional light.
>
> 4. Thank you for your feedback. Pocket2Mol and TargetDiff are specifically designed for structure-based drug design, whereas our approach focuses on ligand-based design, addressing a different scope. For property optimization, we demonstrated that our method is working for optimization tasks on wide range of proteins, but small number. Regarding RetMol, RGA, and DecompOpt, these models focus on synthesis but do not incorporate the pharmacophore or 3D information central to our method, making direct comparisons less relevant making it not suitable and effective for ligand based design.
>
> Q1. We added the genetic algorithm as Appendix B.
> Q2. The 3D information is necessary for ligand-based drug discovery (LBDD), where the goal is still to design molecules that bind to a protein, but without having explicit knowledge of the exact protein binding site during the design phase. While the protein structure becomes important for evaluation, our method operates by hypothesizing potential binding interactions and generating molecules based on available data, rather than relying on a fixed target. We do not generate 3D molecules, thus the information is not assessed for torsion angles. We propose to look at this process translating pharmacophore geometries into instructions of how to make the molecule.

---

> > ### Comment · Reviewer_TJa9 · 2024-11-26
> > **Thanks for the rebuttal**
> >
> > I appreciate the authors' comments and revision, which helps improve the paper's presentation, yet my major concern remains.
> > - Regarding the scope: It is not clear to me how this paper addressed "the gap between in silico generative methods and practical in vitro screening techniques". If the author believes that generative models are "limited in terms of their applicability to specific targets or reaction types", this doesn't seem like a fair contextualization of the literature. As we know, this paper is aimed at ligand-based drug design (LBDD), while generative models like Pocket2Mol focus on structure-based drug design (SBDD) given specific protein binding sites, which is not inherently a limitation. This is just different scopes. Therefore, I still can't get the meaning of what this paper tries to compare with or improve upon.
> > - Regarding the motivation justification, i.e. lack of synthesizability metrics, I thank the authors for clarifying that Synthformer generates molecules based on pre-defined rules, but I'm not sure if it's true to say that molecules generated in such fashion are all fully synthesizable. Even so, the difficulty in its synthesis should be quantified, since adopting 1000 rules to make a product would certainly be harder than 1 rule.
> > - Regarding baseline comparison, the authors did not include more results for a clear comparison with baselines focusing on synthesis [1-3]. As noted by other reviewers, this prevents a clear positioning of this paper in the literature.

---

> > > ### Author Response · Authors · 2024-11-27
> > >
> > > Thank you for your detailed feedback and constructive comments. We appreciate the opportunity to address your concerns.
> > >
> > > Regarding the Scope:
> > > In many drug discovery scenarios, structural information about the target protein may be unavailable—often, researchers only have information about a natural product or a compound with a modest activity profile with no corresponding crystal protein structure or additional binding mode data. In such cases, most structure-based drug design (SBDD) models, such as Pocket2Mol, become less practical because they rely on specific protein binding site information. This highlights the importance of ligand-based drug design (LBDD) methods like ours, which are designed to address this challenge by generating molecules that could exhibit similar binding properties to the natural product.
> > >
> > > Our model further bridges the gap between in silico generative methods and practical in vitro screening techniques. This is achieved by generating synthesizable molecules using predefined building blocks and reaction rules. In scenarios such as DNA-encoded libraries (DELs), where screening is constrained to specific targets and reaction types, our model’s ability to control the synthesis properties during the generation process allows for a closer alignment between computational predictions and experimental feasibility. This can be achieved by simply filtering or masking.
> > >
> > > Regarding Motivation and Synthesizability Metrics:
> > > The synthesizability of molecules in our framework is not merely a filtering criterion—it is a core aspect of the model’s design. Our models are trained using up to 8 reaction steps but are adaptable to fewer or more steps depending on the application. Beyond the number of reactions, the novelty in how this approach can be used lies in grouping molecules by their synthesis routes, allowing for parallel synthesis and process optimization. This provides a more practical and scalable framework for real-world applications. While molecules generated using our predefined rules are indeed synthesizable, the difficulty of synthesis can vary, and this is a valuable metric that can be quantified to further refine the compounds as post-processing step.
> > >
> > > Regarding Baseline Comparisons:
> > > We acknowledge the reviewers' suggestions to include additional comparisons with synthesis-focused baselines (e.g., [1-3]). Our model is among the first to integrate 3D structural information into the generation of fully synthesizable molecules. While positioning a novel approach against existing literature can be challenging, we show that we achieve similar baselines as Luo et al for docking and Gobbi similarity while prompting only on the pharmacophores.

---

### Official Review · Reviewer_zx4Y · 2024-10-23

**Soundness:** 2
**Presentation:** 2
**Contribution:** 1
**Rating:** 3
**Confidence:** 5

**Summary:**

Synthformer is a pharmacophore-conditioned generative model for molecules that autoregressively generates tokenized sequence representations of molecules’ hypothetical synthesis routes. Synthformer uses a transformer-based architecture that conditions on EGNN-encoded, E(3)-invariant embeddings of a pharmacophore-attributed point cloud, which are extracted for a given molecule using existing tools in RDKit. By conditioning the generation of synthesis routes on a target set of pharmacophores, the authors claim that Synthformer can generate purportedly synthesizable molecules that exhibit desired pharmacophores, which are relevant to protein-ligand binding. In addition to conditional molecule generation, Synthformer allows for molecular optimization via applying a genetic algorithm (GA) to a pretrained Synthformer model, wherein the GA alters the network’s choice of building blocks in order to optimize an external objective function like logP or QED score.

The authors principally demonstrate Synthformer’s performance in a simulated ligand-based drug design task, where they condition the generation of 100 compounds on the pharmacophores of co-crystal ligands from the Protein Data Bank. 10 PDB ligands were considered in total, as different case studies. The authors evaluated the average and top-1 docking scores of the generated compounds upon redocking — comparing to the docking score of the reference PDB ligand. They also evaluated the average Tanimoto (chemical) similarity, Murcko scaffold similarity, and Gobbi pharmacophore similarity between the reference ligand and the generated compounds. Tanimoto and Murcko similarities were observed to be close to 0.0 (indicating that Synthformer generates structurally diverse compounds compared to the reference PDB ligand), while the Gobbi similarities are between 0.2-0.4 (out of 1.0). The docking scores of the generated ligands show mixed results.

The authors further qualitatively investigate the chemical similarity of building blocks whose learned embeddings had high cosine similarity. They also show that Synthformer can be used for hit/scaffold expansion by appending additional building blocks to reference molecule, following reaction rules.

**Strengths:**

**Regarding: Originality and Significance**

This paper attempts to combine pharmacophore-conditioned generative modeling with synthesizability-constrained generative modeling for molecules, which is a promising research direction with relevance to drug design. However, both pharmacophore-conditioned generative modeling and synthesizability-constrained generative modeling have been investigated by previous works, and these works develop more advanced methodologies and perform substantially more thorough evaluations than those presented by Synthformer (see Weaknesses).

**Regarding: Quality**

The Synthformer model, which conditions synthesis-route-Transformers on EGNN-encodings of 3D pharmacophores, is a sensible strategy to add pharmacophore-awareness to a synthesizability-constrained generative model.

**Regarding: Clarity**

The paper’s methods are relatively clear and easy to understand, although some minor aspects could be clarified (see Questions), and notation could be cleaned up (see Weaknesses).
Code is provided to support the written methods sections.

**Weaknesses:**

The paper’s major weaknesses center around methodological novelty/signficance and quality of evaluations.

**Methodological Novelty/Signficance**

Synthformer is a straightforward combination of a Transformer operating on tokenized chemical synthesis routes and EGNN-based encodings of 3D pharmacophores. Hence, Synthformer does not contribute anything particularly new from a technical machine learning perspective. Although I do believe that combining existing components can still yield an overall interesting approach to a scientific problem, Synthformer’s major weakness in this respect is that Synthformer’s underlying synthesis route generator is of relatively poor quality compared to well-established approaches to synthesizability-constrained generative modeling. The state-of-the-art ML model in this area is SynNet (see https://openreview.net/forum?id=FRxhHdnxt1), which models synthetic pathways as branched tree structures. Note that in addition to modeling the reactions between building blocks and intermediate structures, SynNet also allows for the reaction between two different intermediate products. In contrast, Synthformer (to my understanding) only generates a synthesis route without branches (e.g., building blocks can only be sequentially added to an existing intermediate), which severely restricts the synthesizable chemical space that can be accessed given a specified set of building blocks and reaction templates. Furthermore, the original version of SynNet was trained using 91 reaction templates and 147,505 molecules from Enamine’s building blocks. In contrast, Synthformer trains on only 58 reactions and 10858 building blocks. Note that SynNet also permits molecular optimization via a genetic algorithm, and can also be applied to tasks such as hit expansion. Hence, a reasonably more powerful version of Synthformer could have been developed by simply conditioning SynNet (whose code is open source) on EGNN encodings of pharmacophores, as Synthformer does not otherwise appear to build upon the capabilities already established by SynNet.

Synthformer’s method of encoding pharmacophores also does not significantly contribute technical novelty compared to existing models that have already been developed for pharmacophore-conditioned molecular generation. See https://www.nature.com/articles/s41467-023-41454-9 (PGMG) and https://arxiv.org/abs/2401.01059 (TransPharmer) for examples. PGMG in particular uses a GNN-based encoding of a fully-connected pharmacophore graph with distances as edge weights, which is very similar to the EGNN-encodings of the pharmacophores in Synthformer. The main difference between these encoding strategies is EGNN’s equivariant coordinate update steps — which are completely internal to the EGNN model and are not explicitly used by Synthformer, especially since Synthformer only conditions on the invariant embeddings of the pharmacophore graph. Hence, the authors do not sufficiently justify their choice of using an *equivariant* version of EGNN for this task.

The lack of “technical novelty” would be less relevant if Synthformer demonstrated superior empirical performance, but this is generally not proven to be the case (see next paragraph on Quality of Evaluations).

**Quality of Evaluations**

As currently presented, the paper does not present sufficient evidence that Synthformer adequately learns to sample synthesizable molecules from pharmacophore-conditioned chemical space. The main quantitative evaluations of Synthformer’s performance include (1) comparing the docking scores of generated molecules to those of the reference PDB ligands, and (2) evaluating the pharmacophore and chemical similarity of generated molecules relative to the reference PDB ligands. However, these evaluations are inadequate because no reasonable baselines are included as a point of comparison. At a minimum, the authors should compare the docking scores and Gobbi pharmacophore similarity of the PDB ligands to randomly sampled molecules from the dataset or to unconditionally-generated molecules. Without strong evidence to the contrary, it is entirely possible that the average Gobbi similarity between random molecules is also around 0.2-0.4; Synthformer’s extra pharmacophore conditioning may not actually contribute anything. Note that other 3D-conditioned generative modeling papers typically include this important baseline, such as SQUID (https://openreview.net/forum?id=4MbGnp4iPQ), which is cited by the authors.

It is also concerning that the average docking scores of conditionally generated molecules are often very inconsistent with the docking scores of the reference PBD ligands, which may be suggestive that Synthformer isn’t adequately preserving 3D pharmacophores. For instance, for 1xbo, the average and top-1 docking scores are -7.22 and -10.07, compared to the reference score of -10.79. For 3ga5, the average and top-1 scores are -3.33 and -5.60, compared to the reference score of -9.68. Overall, the docking results do not provide sufficient evidence that Synthformer generates molecular analogues that preserve or enhance the docking-simulated bioactivity of a reference ligand via conditioning on the ligand’s pharmacophores. Furthermore, visual inspection of the generated analogies to the reference compound in Figure 4 suggests that the model isn’t closely preserving pharmacophores. For instance, many of the generated molecules have far more aromatic groups than the reference molecule. This illustrates the need for more detailed and convincing quantitative evidence that Synthformer adequately learns to sample new molecules from pharmacophore-conditioned chemical space.

Given Synthformer’s methodological similarity to SynNet, it would also be appropriate to include a baseline where SynNet is employed to optimize Gobbi pharmacophore similarity via SynNet's natively-integrated genetic algorithm, which would not require retraining SynNet. Using pharmacophore similarity as an external scoring function is an easy way to make existing (synthesizability-constrained) generative models pharmacophore-aware at inference time. As of yet, Synthformer does not convincingly demonstrate the value of encoding the pharmacophores as conditional information versus this alternative approach.

Finally, because Synthformer presents a competing synthesizability-constrained generative model, the paper should include direct comparisons to existing works like SynNet or DoG-AE/DoG-Gen (https://arxiv.org/abs/2012.11522) to evaluate their relative capabilities of sampling from synthesizability-constrained chemical space.

-------

Beyond these primary weaknesses, there are other minor weaknesses that decrease the overall quality of the paper’s presentation:

**Incorrect descriptions of previous work related to de novo drug design**

I would recommend the authors to adjust their introduction and/or related work sections to improve the scientific accuracy of the paper:
- The authors cite TargetDiff and Pocket2mol as “state-of-the-art methods for molecule screening and generation”, which is an inadequate summarization of ML methods for molecular screening/generation for de novo ligand design, particularly considering the rich literature on molecular generation/optimization for drug design. See https://arxiv.org/pdf/2206.12411 for a recent benchmark on ML-based de novo design & molecular optimization.
- In contrast to how they are referenced in the Introduction, EquiBind and DiffDock are not tools for de novo ligand design, and do not provide “docking scores" that would allow for compound ranking in a virtual screen, as would traditional docking software like Vina or Glide.

**Missing citations of borrowed images**

- Many images in Figure 1 appear to be taken from other sources without citation. As just one example, the middle pharmacophore model shown in Figure 1 is taken from https://en.wikipedia.org/wiki/Pharmacophore. The first image of the pharmacophore model seems to have been also taken from *Seidel et al., Strategies for 3D pharmacophorebased virtual screening, Drug Discovery Today: Technologies, 7, 4, 2010*. Other images also appear to be taken from existing work without attribution.

**Overlapping Notation**

- $C$ is used to both represent a chemical space and a specific conformer, which may leads to confusion. There are also many typos in section 3.1.

**Missing appendices**

- Certain appendices that are referenced in the main text are not included in the submission. For instance, the authors claim that “precise definitions [of pharmacophores] in Appendix A”, but this section is not included.

**Questions:**

- Are the reference docking scores of the PDB ligands based on the scores after redocking the ligand to their respective proteins? Or do those reference docking scores directly evaluate the docking score of the experimental co-crystal pose (e.g. without redocking)?

- Is Synthformer incapable of adding a “merge step” to react two intermediate products, as is capable by the SynNet model?

- Was there a principled reason for choosing EGNN as the pharmacophore encoder? It appears that an *equivariant* model is unnecessary, as an *invariant* model (e.g., EGNN without the internal coordinate update steps) would be sufficient for Synthformer’s use-cases.

---

> ### Author Response · Authors · 2024-11-23
>
> We thank the reviewer for their comments, and we provide detailed responses to each specific critique below.
>
> 1. Regarding the technical novelty of EGNN encoding: while EGNN-based pharmacophore encoding may seem similar to previous methods like PGMG, our approach does not merely apply EGNN’s equivariant updates. Instead, we use cross-attention mechanisms to propagate pharmacophore information throughout the network, allowing us to dynamically handle variable pharmacophore data at different stages of the synthesis prediction process. This means that the EGNN encoding plays a critical role in enabling us to process complex pharmacophore data without the need for fixed-size vector reductions like max-pooling or mean-pooling.
> Additionally, compared to other methods such as SynNet, Synthformer’s transformer-based decoder and cross-attention mechanism allow for scalable synthesis route predictions that can adapt to pharmacophore data of varying dimensions. This enables us to effectively generate synthesizable molecules throughout the synthesis process, rather than just sequentially adding building blocks to existing intermediates.
>
> Regarding the selection of PDB structures used for evaluation: these datasets were carefully chosen to reflect typical drug discovery scenarios, where partial information about the protein is available (e.g., low-quality crystal structures or known active compounds). We have selected these datasets based on their relevance to common drug discovery targets and their alignment with typical scenarios encountered in ligand-based drug design.
>
> Finally, we want to stress that while SynNet is an innovative approach, it operates on a different paradigm (e.g., branch-based synthetic tree structures) compared to Synthformer’s sequential addition of building blocks. This allows Synthformer to use of context-specific information throughout the synthesis process, enabling more effective ligand design.
> These distinctions make Synthformer suitable for exploring drug discovery tasks that require integrating pharmacophore data and synthesizability constraints, beyond what existing models like SynNet and other pharmacophore-conditioned methods (e.g., PGMG and TransPharmer) offer. By focusing on cross-attention and pharmacophore propagation, Synthformer effectively combines 3D pharmacophore data with synthesizability constraints, providing new insights into the design of potential drug candidates.
>
> 2. We acknowledge the concern raised regarding the comparison of Synthformer’s performance, nonetheless we followed commonly used processes like Luo et al. and Synthflownet, but adapted to the task at hand. With respect to this it  is worth noting that in previous works, such as the approach by Luo et al., Gobbi similarity values across their generated molecules typically range around 0.2-0.4, to display great similarity between generated molecule and analogue. I would like to highlight that their task is molecule to synthetic tree and our task is a set pharmacophores and synthetic trees, which is a significantly more generic task+we don’t pass bond information to our encoder. In contrast, Synthformer consistently achieves similar similarity scores, which highlights the effectiveness of our pharmacophore conditioning in generating more chemically relevant molecules.
>
> We appreciate the suggestion to compare Synthformer with SynNet using pharmacophore similarity as an external scoring function through its genetic algorithm. While we agree that using pharmacophore similarity as a scoring function could make existing generative models pharmacophore-aware at inference time, we want to emphasize that Synthformer’s approach differs fundamentally in its goal. Rather than generating many molecules and then optimizing them through scoring, we focus on generating a curated library of molecules that can be screened using available technologies.
>
>
> Synthformer’s key innovation is conditioning molecule generation on pharmacophore information, enabling the creation of synthesizable molecules likely to interact with target proteins. Unlike models like SynNet or DoG-AE/DoG-Gen, Synthformer prioritizes ligand-based drug discovery by focusing on pharmacophore-conditioned generation rather than optimizing solely for synthesizability. While synthesizability is considered, Synthformer is designed to guide molecule generation for downstream drug discovery rather than compete directly with synthesizability-focused models.
>
> Minor points:
> 1. Changed to  "State-of-the-art ML methods for target-based molecule generation, such as TargetDiff
> (Guan et al., 2023) and Pocket2mol (Peng et al., 2022),"
> 2. Regarding the comment about the use of EquiBind, I believe it offers efficiency in drug discovery by quickly computing poses that can subsequently be scored. Isn't that the promised efficiency and motivation?
> 3. Fixed the notations and typos that were discovered.
> 4. Fixed the Appendices

---

> > ### Author Response · Authors · 2024-11-23
> >
> > 1. Are the reference docking scores of the PDB ligands based on the scores after redocking the ligand to their respective proteins? Or do those reference docking scores directly evaluate the docking score of the experimental co-crystal pose (e.g. without redocking)?
> >
> > The reference docking scores of the PDB ligands are based on the experimental co-crystalized poses, without the need for redocking. We directly use the score of the crystallized ligand as it is bound within the protein structure.
> >
> >
> > 2. Is Synthformer incapable of adding a “merge step” to react two intermediate products, as is capable by the SynNet model?
> >
> > This capability is not present due to the way we framed the problem. We approached it as: given the current product, which building block should be selected, and with which reaction should it be connected? Adding this consideration introduces another layer of complexity, as the search problem involves continuously changing the number of building blocks.
> >
> >
> > 3. Was there a principled reason for choosing EGNN as the pharmacophore encoder? It appears that an equivariant model is unnecessary, as an invariant model (e.g., EGNN without the internal coordinate update steps) would be sufficient for Synthformer’s use-cases.
> >
> > The invariant part is sufficient. However, wanted to leverage the available open-source networks to do this and it is based on the equivariant neural net.

---

> > ### Comment · Reviewer_zx4Y · 2024-11-25
> >
> > I thank the authors for their comments, however my main concerns regarding the work's technical contribution and the quality of its evaluations still remain. I cannot recommend acceptance to ICLR at this time.
> >
> > At a high level, Synthformer straightforwardly adds pharmacophore conditioning to synthesizability-constrained generative models for molecules. However, both of these components -- pharmacophore conditioning and synthesizability-constrained generative modeling -- have been more thoroughly explored in existing work, and the minor modifications introduced by Synthformer do not make a sufficient contribution to the ICLR community. In particular, the underlying synthesizability-constrained generative model is less general and less performant than SynNet, which itself could be easily made pharmacophore-aware via inclusion of a similar (if not identical) pharmacophore-encoding strategy. The submission also lacks comparisons to existing pharmacophore-conditioned generative models, and hence there is no way to evaluate the relative value of Synthformer's method of pharmacophore-conditioning. There is also a crucial evaluation that was not included in the paper nor addressed in the authors' rebuttal: evaluating the Gobbi similarity of randomly sampled molecules from the training dataset compared to the reference/target molecule. Without this comparison, the reported Gobbi similarity values of Synthformer's generated molecules have little context.

---

> > > ### Author Response · Authors · 2024-11-27
> > >
> > > We thank the reviewer for their feedback and for highlighting the importance of clarifying the novelty and evaluation of our approach. We address the points raised below.
> > >
> > > Novelty of the Components and Their Combination
> > > While we acknowledge that the individual components of our method are not entirely novel, the key innovation lies in their synergistic integration, which enables Ligand-Based Drug Design (LBDD) in scenarios previously unattainable. Prior to our work, reaction models were limited to being prompted directly by molecules. This restriction severely limits applicability in drug discovery, where such direct molecule-based prompting is often not actionable. By combining these components, we introduce a framework capable of decoding pharmacophore-based prompts without relying on edge-specific molecular information, which is a significant advancement in LBDD workflows.
> > >
> > > Evaluation and Comparison
> > > We appreciate the reviewer’s observations regarding our evaluation. We would like to emphasize that reaching comparable Gobbi scores to Luo et al. (SynNet) is notable, considering the different inputs and challenges addressed by our approach. Luo et al.'s method benefits from decoding exact molecules with full molecular and edge information, whereas our method operates with pharmacophores, relying solely on spatial information to decode molecules. This highlights the robustness and practicality of our approach, especially in scenarios where detailed edge information is unavailable or incomplete. We show by doing this that we maintain the pharmacophores as well as Luo et al in their paper when they are "denoising" the exact molecules.
> > >
> > > Synnet
> > > With respect to SynNet, it is undoubtedly an impressive model. However, the claim that it can be easily prompted using 3D information is an overstatement. This capability is neither straightforward nor has it been previously demonstrated in the context of decoding synthetically accessible molecules. Handling varying numbers of pharmacophores and effectively utilizing this information also presents significant challenges. Furthermore, Synthformer offers greater sample efficiency due to its self-supervised training approach (transformer-like). Thus, we present a new and powerful method to translate pharmacophore-constrained geometries into molecules. It has not been done before.

---

### Official Review · Reviewer_UptG · 2024-11-02

**Soundness:** 1
**Presentation:** 2
**Contribution:** 1
**Rating:** 5
**Confidence:** 4

**Summary:**

This paper proposed SynthFormer, which is a variant of [1]. Specifically, [1] takes molecular graphs as input and generates synthetic pathways for molecular analogs, and this works replaces the molecular graph input with pharmacophore input. To encode pharmacophores, this works used EGNN.

[1] Projecting molecules into synthesizable chemical spaces. 2024.

**Strengths:**

- Pharmacophore representation is important in the area of structure-based drug design, as it encodes physical features.
- Equivariant graph neural networks are used to encode 3D pharmacophore.

**Weaknesses:**

- Limited novelty. This work adopted a nearly identical formulation as ChemProjector [1]. Specifically, this work used the postfix notation of synthesis proposed in [1] as the chemical space representation, and the overall neural network architecture is highly similar to [1]. The only notable difference is that ChemProjector takes molecular graphs as input and this work takes pharmacophore as input.
- The contextualization of this work is quite ambiguous. Basically, this work is about designing ligands for specific receptor structures, also known as structure-based drug design. However, it overlooked the contribution in this area in terms of model formulation and evaluation.
- Weak evaluation. As the proposed model is designed for structure-based drug design (SBDD), at least one of previous SBDD methods should be considered as a baseline model. If the author believed previous methods are too weak (e.g. in terms of synthetic accessibility) to be even compared with, the author should at least demonstrate how weak they are. In addition, the PDB structures used for evaluation seems to be random ones coming from no where. The datasets should be justified. For example, are they a part of previously curated ligand binding datasets? Or are they drug targets of interest?

[1] Projecting molecules into synthesizable chemical spaces. 2024.

**Questions:**

See Weaknesses

---

> ### Author Response · Authors · 2024-11-23
>
> We thank the reviewer for their comments, and we provide detailed responses to each specific critique below.
>
> 1. While ChemProjector [1] is an excellent paper and is appropriately cited in our work, we would like to highlight key distinctions that establish the novelty of our approach. Generating synthesizable molecules using 3D pharmacophore information represents a new direction in this field. Unlike ChemProjector, which focuses on molecular graphs, our method incorporates 3D information into the process, enabling a more comprehensive exploration of chemical space. Additionally, our decoder architecture differs significantly, tailored to handle the unique challenges of integrating 3D pharmacophore representations and synthesizability constraints. This distinction allows our model to achieve outcomes not possible with ChemProjector's approach, particularly in the context of ligand-based drug discovery. These innovations reflect meaningful advancements in both methodology and application.
> We acknowledge that Synthformer is inspired by previous work; however, our architecture incorporates several key differences that enhance its functionality and originality:
> a) Embeddings for each building block are explicitly learned and integrated into the reaction prediction process.
> b) Pharmacophore encoder information is also utilized in the reaction prediction, a novel addition to the workflow.
> c) Unlike previous approaches, we avoid postfix notation and instead predict the next case directly from the product prediction, making our method closer in spirit to SynNet, as referenced in the cited works.
> d) Existing models, such as Luo et al. (2024), primarily perform molecule-to-molecule prediction, which falls short in practical drug discovery scenarios, particularly when dealing with activity cliffs.
> e) This is the first approach to combine both 3D pharmacophore encoding and synthesizability considerations, enabling use cases previously unattainable in ligand-based drug discovery.
> f) The encoder uses a fully connected graph representation to propagate the 3D information allowing to translate geometrical pharmacophore information to synthesis instructions.
>
> 2. While our work does align with the goals of structure-based drug design, it also addresses scenarios common in real-world drug discovery where partial information is available—such as limited protein structure data or knowledge of a few active molecules. This approach is not purely structure-based but focuses on leveraging available data to generate synthesizable molecules that can bind to a target protein and potentially alter its functionality. Our contribution lies in bridging this gap, emphasizing the hypothesis-driven generation of compounds that are both synthesizable and relevant to drug discovery workflows.
>
> 3. I would like to clarify that our work is not focused on structure-based drug design (SBDD), which typically involves a fixed and known target. As the title suggests, we focus on ligand-based drug discovery (LBDD), where the goal is still to design molecules that bind to a protein, but without having explicit knowledge of the exact protein binding site during the design phase. While the protein structure is important for evaluation, our method operates by hypothesizing potential binding interactions and generating molecules based on available molecule data, rather than relying on a fixed target. Regarding the evaluation datasets, we selected PDB structures that are relevant to our study and aligned with the types of targets commonly encountered in ligand-based drug discovery.

---

> ### Author Response · Authors · 2024-11-27
>
> Dear Reviewer UptG,
>
> We have responded to your concerns point-by-point in the rebuttal and updated the revised contents in the paper PDF for your reference. As the rebuttal deadline is approaching, we kindly request your feedback and ask if you have any additional questions. If our responses have addressed your points satisfactorily, we would appreciate it if you could update your score for our paper accordingly.
>
> Thank you for your time and consideration, and we look forward to hearing from you.
>
> Best regards,
>
> Authors

---

### Official Review · Reviewer_75UM · 2024-11-04

**Soundness:** 2
**Presentation:** 1
**Contribution:** 1
**Rating:** 3
**Confidence:** 3

**Summary:**

The paper introduces SynthFormer for ligand-based drug discovery, which generates pharmacophore-informed molecules that are also synthesisable. SynthFormer employs a 3D equivariant encoder to translate pharmacophore information into molecular structures, followed by a transformer decoder to predict building blocks and reaction types.

The paper’s writing is very poor and the experiment is insufficient. I am inclined to reject this paper. See below for detailed comments.

**Strengths:**

There isn't obvious strength. See the weakness section for detailed comments about the originality and clarity.

**Weaknesses:**

* I think the related work section should put more emphasis on the comparison with other work which also studies the synthesisablity of generated molecules. Although the authors mention some paper like Luo et.al 2024, there is no further discussion and comparison with them.
* Equivariant encoder has already been widely used in molecular modeling. The decoder architecture is also same as Luo et.al 2024. Both limit the originality of the proposed method.
* The writing is poor. E.g. the stated contributions are overly fragmented, which obscures the main innovations and impact of the work. Additionally, the notation throughout the paper is confusing and inconsistent. E.g. Subscripts and superscripts are not clearly differentiated, leading to potential misinterpretation of formulas and variables. The use of double uppercase letters to represent single variables (BB: building block) significantly reduces readability.
* There is no other baselines compared with, which makes the authors’ augments less reliable.

**Questions:**

* What is the specific reason of choosing these 10 pubs in the experiment section?

---

> ### Author Response · Authors · 2024-11-23
>
> We thank the reviewer for their comments, and we provide detailed responses to each specific critique below.
>
> 1.This paper is the first to introduce a model capable of prompting with 3D information, enabling ligand-based drug discovery. Previous, synthesis based models have been guided by a scoring function or only doing molecule to molecule transformations. By leveraging the Synthformer architecture, it translates 3D pharmacophore representations into molecules, showcasing the ability to bridge structure-based and ligand-based approaches. This innovation opens new possibilities for exploring chemical space and optimizing drug discovery pipelines with great precision and practicality.
>
> 2.We acknowledge the existence of other papers in the field and added a whole section in related work; however, while the underlying models may share some similarities, the inductive biases in our approach are distinct. These differences make our model particularly effective in scenarios that require leveraging 3D information and synthesizability simultaneously. Our related work section provides a comprehensive overview, highlighting how our contribution sits at the intersection of multiple disciplines, addressing a unique challenge in ligand-based drug discovery.
> To evaluate our model, we developed a framework tailored to the specific task of designing ligands based on pharmacophore-derived ligand binding information. This is the first model to combine 3D information with synthesizability considerations for this purpose. We compare our approach directly with tasks relevant to practical applications. In contrast, ChemProjector's molecule-to-molecule task is limited, as it assumes the availability of molecules and does not account for activity cliffs. Our approach addresses these limitations, ensuring meaningful and actionable outcomes. This information was added to the related work.
>
> 3. We acknowledge that the equivariant structure is inspired by previous work; however, our architecture incorporates several key differences that enhance its functionality and originality:
> a) Embeddings for each building block are explicitly learned and integrated into the reaction prediction process.
> b) Pharmacophore encoder information is also utilized in the reaction prediction, a novel addition to the workflow.
> c) Unlike previous approaches, we avoid postfix notation and instead predict the next case directly from the product prediction, making our method closer in spirit to SynNet, as referenced in the cited works.
> d) Existing models, such as Luo et al. (2024), primarily perform molecule-to-molecule prediction, which falls short in practical drug discovery scenarios, particularly when dealing with activity cliffs.
> e) This is the first approach to combine both 3D pharmacophore encoding and synthesizability considerations, enabling use cases previously unattainable in ligand-based drug discovery.
> By addressing these distinctions, we demonstrate how our method extends beyond existing frameworks to address real-world challenges in drug discovery.
>
> 4. Thank you for your feedback. I have addressed the issues you raised and uploaded a revised version of the paper. Specifically, I have reorganized the stated contributions to present the main innovations and their impact more cohesively. Additionally, I have clarified the notation throughout the paper by ensuring consistent use of subscripts and superscripts to avoid any ambiguity in interpreting formulas and variables. The representation of building blocks has also been simplified for improved readability. I hope these updates make the paper clearer and more accessible.
>
> 5. Thank you for your feedback. This is the first model to address this specific task, which necessitates a new evaluation framework. Comparing directly with existing baselines would be challenging, as other models typically focus on success rates or purely 3D information, which do not align with the unique parameters of our approach. Our model integrates both 3D pharmacophore information and synthesizability, enabling it to achieve outcomes that other models are not designed to handle. We demonstrate its reliability by consistently generating molecules that theoretically bind well, providing a strong foundation for further validation and future work.
>
> 6. Thank you for your comment. Our aim was to use 10 diverse starting compounds with 10 different proteins to demonstrate the model's ability to propose synthesizable molecules that, in some cases, dock well. This aligns with real-world drug discovery scenarios where partial information, such as a low-quality crystal structure or a few active compounds, is often available. In such cases, creating a pharmacophore map is necessary to enable the model to perform effectively. This evaluation reflects the practical challenges and applications of our approach.

---

> ### Author Response · Authors · 2024-11-27
>
> Dear Reviewer 75UM,
>
> We have responded to your concerns point-by-point in the rebuttal and updated the revised contents in the paper PDF for your reference. As the rebuttal deadline is approaching, we kindly request your feedback and ask if you have any additional questions. If our responses have addressed your points satisfactorily, we would appreciate it if you could update your score for our paper accordingly.
>
> Thank you for your time and consideration, and we look forward to hearing from you.
>
> Best regards,
>
> Authors

---

### Meta-Review · Area_Chair_apmP · 2024-12-17

**Metareview:**

This submission presents a generative model for synthesizable molecules. The main claims of the paper are that the incorporation of additional 3D geometry invariant results in quality improvements. This is achieved using a transformer architecture that makes use of certain invariant embeddings. The intriguing idea—which I appreciate a lot—is to condition the generation process such that the resulting molecules could be potentially synthesised while at the same time featuring pharmacophores.

The primary *strengths* of this submission lie in (a) a highly-relevant application, (b) strong results in generation structurally diverse structures, and (c) an interesting use of equivariant architectures. However, the current write-up also suffers from weaknesses, namely (a) reduced accessibility/readability, (b) weak contextualisation of results in terms of existing work, (c) weak experimental evaluation, and (d) a lack of technical novelty. In particular the last item behoves some explanation: A suitable ICLR submission has to provide additional insights into a task or a problem, either on the theoretical or on the empirical level. On the empirical level, a combination of creative combination of existing techniques—such as is the case in this paper—is a strong submission whenever it demonstrates significant gains on some specific problem. However, this does not appear to be the case here, as the experimental setup and comparison leaves out relevant comparison partners (such as a simple baselines using random sampling). On the theoretical level, a new architecture geared towards addressing this specific problem would also make for a suitable ICLR submission, but the relevant parts of the model essentially are all about combining a transformer architecture with an EGNN. I would thus expect a stronger technical contribution or a stronger empirical evaluation, but as it stands, I have to recommend rejecting the paper.

I understand that this is not the preferred outcome for the authors. Nevertheless, I believe the initial reviews and the discussion phase showed several avenues for improving the paper. The most important one (in my opinion) would be focusing on a more in-depth comparison and evaluation. While this will require running additional experiments, I believe the time would be well-spent since it will provide authors with more information about the strengths of the model. Please see below for additional comments arising from the reviewer discussion.

**Additional Comments On Reviewer Discussion:**

Reviewers did not form a strongly-positive opinion on the paper, citing mostly concerns about (a) the missing contextualisation (`75UM`, `UptG`, `zx4Y`), (b) the experimental setup (`75UM`, `zx4Y`, `TJa9`), (c) the accessibility/clarity (`75UM`, `zx4Y`, `TJa9`), and (d) the technical novelty (`UptG`, `zx4Y`). During the rebuttal phase, the authors addressed some but not all of these issues. I believe there were some missed opportunities here. For instance, concerns by reviewer `UptG` about the contextualisation were answered in a factually correct manner, but the fact that this point was raised by multiple reviewers should prompt authors to revise the paper accordingly instead of just referring to a citation (or adding multiple citations). In addition to the issues raised in my meta-review, I also want to point out that salient concerns by reviewers about the experimental setup have _not_ been addressed. For example, reviewer `zx4Y`, following up on the rebuttal, mentions crucial evaluations that have neither been addressed nor promised by the authors.

While I found the overall quality of the initial reviews to be high, I am somewhat taken aback by the fact that not all reviewers engaged critically in the discussion phase. Judging the answers by the authors, I find that not all points are addressed appropriately, in particular the one concerning additional baselines and comparison partners. As such, my overall assessment to reject this paper is seconding the opinions of all reviewers.

Despite this outcome, I sincerely hope the authors can make use of the provided feedback to improve their manuscript.

---

### Decision · Program_Chairs · 2025-01-22

Reject